# Deep Learning Approaches for Prognosis of Automated Skin Disease

**DOI:** 10.3390/life12030426

**Published:** 2022-03-15

**Authors:** Pravin R. Kshirsagar, Hariprasath Manoharan, S. Shitharth, Abdulrhman M. Alshareef, Nabeel Albishry, Praveen Kumar Balachandran

**Affiliations:** 1Department of Artificial Intelligence, G.H. Raisoni College of Engineering, Nagpur 412207, India; pravinrk88@yahoo.com; 2Department of Electronics and Communication Engineering, Panimalar Institute of Technology, Poonamallee, Chennai 600123, India; hari13prasath@gmail.com; 3Department of Computer Science & Engineering, Kebri Dehar University, Kebri Dahar P.O. Box 250, Ethiopia; shitharth.it@gmail.com; 4Department of Information Systems, Faculty of Computing and Information Technology, King Abdulaziz University, Jeddah 21589, Saudi Arabia; amralshareef@kau.edu.sa; 5Department of Information Technology, Faculty of Computing and Information Technology, King Abdulaziz University, Jeddah 21589, Saudi Arabia; nalbishry@kau.edu.sa; 6Department of Electrical and Electronics Engineering, Vardhaman College of Engineering, Hyderabad 501218, India

**Keywords:** deep learning, learning algorithms, skin disease, MobileNetV2, LSTM

## Abstract

Skin problems are among the most common ailments on Earth. Despite its popularity, assessing it is not easy because of the complexities in skin tones, hair colors, and hairstyles. Skin disorders provide a significant public health risk across the globe. They become dangerous when they enter the invasive phase. Dermatological illnesses are a significant concern for the medical community. Because of increased pollution and poor diet, the number of individuals with skin disorders is on the rise at an alarming rate. People often overlook the early signs of skin illness. The current approach for diagnosing and treating skin conditions relies on a biopsy process examined and administered by physicians. Human assessment can be avoided with a hybrid technique, thus providing hopeful findings on time. Approaches to a thorough investigation indicate that deep learning methods might be used to construct frameworks capable of identifying diverse skin conditions. Skin and non-skin tissue must be distinguished to detect skin diseases. This research developed a skin disease classification system using MobileNetV2 and LSTM. For this system, accuracy in skin disease forecasting is the primary aim while ensuring excellent efficiency in storing complete state information for exact forecasts.

## 1. Introduction

The human body’s primary feature is the skin. The epidermis, dermis, and subcutaneous tissues comprise it. The skin senses the external environment and protects our inside organs and tissues from harmful microorganisms, pollution, and sun exposure [1]. Numerous environmental and internal variables may affect the skin. Simulated skin damage, toxic exposure, embryogenic infections, a person’s immune function, and genetic problems are all elements that contribute to the development of diseases of the skin. Skin disorders are a significant impact on one’s life and the well. Occasionally, people try to resolve their skin problems via home remedies. If these methods are not appropriate for that skin disease, they may have negative consequences. Because skin disorders are readily transmitted from individual to individual, they must be preliminarily managed. Typically, assumptions about a patient’s condition are drawn based on the physician’s experience and subjective judgments. If the decision is incorrect or delayed, it may have a detrimental effect on health [2]. As a result, this becomes essential and critical to establish effective methods for identifying and treating skin disorders.

Technological development has allowed for the structuring and execution of a skin observation early years foundation identification of skin problems. Numerous improvements are available for the image- and pattern-based detection of various skin disorders. Deep learning is one of the fields that can significantly contribute to the operational and precise detection of multiple types of skin disorders. Infections may be diagnosed using image classification and deep learning [3]. Image classification is a fundamental problem in which many objective categories are defined, and a trained model to recognize each type is developed.

Skin diseases are often overlooked and accorded little attention in their early stages. Individuals’ awareness may contribute to the development of skin cancer. Currently, the skin irritation condition is detected only via biopsies at a future date [4]. The examination is performed manually, taking into account various histological characteristics. Consequently, given that this procedure is carried out manually, it may result in a human mistake; the biopsy results are provided within 1–2 days.

Additionally, the doctor has difficulty determining the kind of skin illness and its stage during the evaluation phase [5]. As a result, prescribing medication becomes more complicated. This issue may be solved by evaluating the microscopic picture using deep learning methods. This suggests that deep learning-based methods may be helpful in quickly identifying clinical information and giving findings. The multifaceted nature of skin disorders, the scarcity and misallocation of competent physicians, and the critical requirement for fast and accurate identification necessitate information treatment. Lasers and photonics-based medical technology advancements have enabled far more rapid and precise skin disorders diagnosis. However, the expense of such diagnostics remains prohibitively high. Deep learning algorithms effectively classify pictures and information [6].

There has been a request in health diagnosis for the accurate recognition of abnormalities and categorization of diseases using X-ray, magnetic MRI, CT, PET, and signal data such as the ECG, EEG, and EMG. The accurate classification of diseases will aid in providing patients with better care. DL methods can resolve important issues by automatically detecting data input characteristics, and they are adaptive to changes in the problem space [7]. With even minimal computational models, deep learning methods will obtain the assumed information to discover and investigate the characteristics in the uncovered patterns in data, resulting in substantial effectiveness. This prompted the researchers to explore using a DL model for classification, in which skin disease classification is based on the picture of the afflicted area suggested work. The study’s primary goal is to provide a state-of-the-art method, called MobileNet V2 with LSTM element, for the objective of accurately classifying skin diseases using a picture taken from a device. Its actual use is to develop an app that captures a view of the afflicted area of the skin to classify it [8]. The scientists chose the MobileNet V2 model because it is computationally efficient and performs well with limited quality pictures. LSTM also effectively manages the potential vanishing problem during repetitions in neural networks, which aids in quicker training of the model [9]. The suggested approach would help medical professionals and patients perform an efficient invasive illness assessment at the lowest feasible price and minimum possible workforce.

### 1.1. Literature Review

Researchers [1] have incorporated numerous methods for recognizing and classifying skin disorders that have been automated. Most diagnostic techniques depend on machine vision, although epidermis identification of these skin disorders does not need radiological imaging. They can identify the condition using image processing methods such as picture modification, equalization, enrichment, feature extraction, and classification combined with traditional images. The skin images collected for illness classification and identification are handled and fed into advanced automation strategies such as ML, DL, ANN, CNN, and back propagation neural networks, as well as classification methods such as SVM and NB classification methods for the prognostication of skincare product illnesses [10,11,12,13,14,15,16,17,18,19,20,21]. Skin disorders may also be identified using image processing techniques such as mathematical morphology for texture analysis [2]. Morphology openings, closure, dilatation, and erosion are primarily dependent on the digital picture produced by thresholding, and, as a consequence, the optimal threshold value must be determined with care. Morphological-based procedures may be insufficient for predicting the development of the injured section depending on the surface of the picture. A genetic algorithm (GA) has developed a method for classifying skin diseases.

The genetic algorithm does have drawbacks, such as taking an extraordinary amount of time to converge on the answer. The system always provides the optimal global solution if such a solution does not lead to an acceptable conclusion [22,23,24,25,26,27]. It has been systematically investigated whether methods such as dermoscopy images trends and lesion assessment using deep convolutional neural networks, image classification techniques, and information enhancement are automatically identified [3]. In addition, [4] has created a computer-based method for analyzing images obtained through ELM. To obtain a binary mask of a skin lesion, standard segmentation methods are combined with a fusing approach. Its form and radioactive characteristics determine the aggressiveness of a tumor. Global and regional factors are also taken into account for improved outcomes. The method significantly increases the rate of earlier detection of metastatic melanoma [4]. In addition, this approach can indicate the use of different algorithms for the classification and prediction of benchmark datasets and a real-time dataset that are helpful in emerging fields and elaborate the use of hybrid artificial intelligence along with optimization techniques for the classification and prediction of various datasets with high accuracy [7]. The algorithms used in various research have been practical in cybersecurity, mobile computing, and cloud computing for more accurate results with different evaluation parameters.

A novel mathematical method for determining a tumor border has been examined [2]. The technique analyzes brightness levels across a path equal to a pattern at each location. In [5], Raman spectroscopy was used to classify a skin lesion in vitro. They conducted their research using a nonlinear neural network classifier. Distinctive patterns in the spectra indicate the presence of specified proteins and lipids, which aids in diagnosing skin diseases. In [6], the authors evaluated skin malignancies and ulcers using electric bioimpedance. Skin cancer and harmless vermin were distinguished using a multi-frequency impedance spectrum. In [7], researchers developed a new recurrent probabilistic area method for segmenting macroscopic lesion areas. Essentially, this method performs a stochastic zone combining at the pixels and then at the national and regional level until resolution. In [6], the focus was on integrating several forecasting models using ensemble methods to improve the accuracy of skin disease categorization. Ensemble methods suffer from imbalanced datasets, and they do not perform well when there are unexpected differences between the considered samples and populations. The categorization of skin disorders using a deep neural network model has shown fantastic performance. Nonetheless, empirical work has shown that the paradigm is unsuitable for pictures with many lesions. Deep neural network designs require a significant level of practice to achieve accuracy, demanding additional computing effort. These designs use a cross-correlation-based approach for the extraction of features categorization, in which internal optical consistency is used to evaluate both time and spectral characteristics while selecting features. Cross-correlation systems are resistant to noise. As a result, the forecasts become more precise. Furthermore, operating in the spectral domain requires a significant commitment to setting up the research and obtaining the findings.

### 1.2. Research Gap and Design Parameters

An automatic skin disease diagnostic system can predict skin illnesses with such a maximum throughput in a short amount of time. When skin disorders are identified early, skin condition disorders such as skin cancer may be avoided. This section addresses the design parameters of a combination method of detecting skin conditions. In reality, source pictures are not uniform in size, structure, height, color, and intensity, among other characteristics. Additionally, good picture separation is required for skin disease prediction [10]. Cropping picture segments for the extraction of features may result in a decrease in the illness classifier. Certain specific criteria are needed to develop an effective hybrid digital skin condition detection mechanism. The four fundamental criteria include, amongst many others, resilience, data segmentation, data retrieval, and prediction. These basic needs are shown in Figure 1. Robustness refers to the ability to identify the damaged skin picture after it has been altered by many typical images processing procedures, including resizing, resizing, translational, wavelet transform, rotations, color mappings, distortion, and compression.

An additional need for a hybrid skin disease detection scheme is information segmentation, which divides the impacted picture region for skin condition forecast. A segment divides pictures into a small number of areas (preliminary classification) or a large number of sections (excessive distribution) (over-segmentation) [11]. The data extract aims to reveal the characteristics produced by the separated picture section. These derived characteristics are utilized to categorize skin disorders effectively. Standardization is another critical requirement for an automatic method of detecting skin diseases [12]. Skin disease prediction is accomplished via the use of deep learning algorithms. Regular, mild, questionable, and melanoma skin disorders are anticipated.

## 2. System Model

Researchers evaluate a series of pictures received first from a client in our suggested system and conduct processing and identification on each image [13]. Then, each print is subjected to a pattern extractor to obtain elements that may be utilized to build a classifier. With this classification algorithm in place, the program can estimate the illness for a fresh picture of a skin condition received through an Android mobile application [14]. Moreover, depending on this projected illness, the system asks the customer a series of questions and, judging by the response, will determine the kind of the disease. Ultimately, our method recommends drug treatments or advice depending on the anticipated outcome of skin illness. We consider three conditions in this framework: dermatitis, ringworm, and histamine. Figure 2 depicts the proposed program’s design, which shows the proposed system’s significant operations. We explore the suggested methods in depth in this section. Image preprocessing is a critical stage in the detection method since it allows for noise removals such as hairs, clothes, and other artifacts while also improving the characteristics of the original image [16]. The primary goal of this phase is to improve the quality of the epidermal picture by eliminating irrelevant and redundant components from the image’s backgrounds for further treatment. A well-chosen set of preprocessing techniques may significantly enhance the service’s efficiency [17].

The preprocessing frame’s goal may be accomplished in three phases: picture improvement, image reconstruction, and laser treatments. Segmentation is the process of determining the form and size of an image’s border. It distinguishes the item from its backdrop using several picture characteristics [18]. After eliminating the sound and hairs from the lesion area, the lesion must be isolated from the surrounding skin. Thus, the diagnostic examination is performed only on the binding site [19]. Many segmentation techniques may be used in this research.

### 2.1. Motivation of Proposed Work: A Design Plan

Skin illnesses include those that begin within the organ and develop on the skin or those that start and appear. This section will analyze these illnesses [25]. Additionally, it describes the topology of an artificial neural network used to diagnose skin disorders. A professional diagnoses skin disorders via the collection of patient information and symptoms. This collection of documented skin diseases and complaints is then extended into numerous logical indicators.

Additionally, the symptoms are compared with the data previously held by the expert system. If a result is made, the physician suggests the illness as a potential cause of skin disease. In some instances, the medical specialist may refer the patient for further lab testing to determine the etiology of the skin disease. The test may be performed to confirm that the illness being evaluated is caused by an organism, such as a bacterium, parasite, virus, or fungus [26]. When a specialist is unskilled or has never seen such a skin disease, he diagnoses via trial and error. This is accomplished by combining all potential circumstances, contrasting them to known ones, and thus limiting the decision. Learning is considered to have occurred throughout this phase if the skin problem is correctly diagnosed and managed. Therefore, the professional relies on their expertise and assessing the client’s complaints.

The steps in this study are shown in Figure 3 [27].

### 2.2. Dermatological Disease Analyses of Diagnostic Systems

Figure 3 illustrates the illness assessment procedure, which comprises the primary gathering and cataloging of skin disorders researched and utilized in the design process. This involves the filing of a complaint, observation, and laboratory test. The illness assessment entails gathering and cataloging all skin disorders that will be researched and utilized in constructing the system’s deep neural network. Figure 3 illustrates various skin disease outbreaks.

Similarly, Table 1 lists skin disorders and their associated symptoms [28]. The National Center for Skin Illnesses in Malaysia was chosen for skin categorization in this research since it utilizes state-of-the-art technology to help diagnose. It employs complexion devices that enable accurate skin color variation classification caused by illness using representative humanoid beings [29]. Additionally, the computer can monitor an individual skin lesion more accurately than human perception, which may be unpredictable. Additionally, the center offers internet sample tests via their internet consultant, who uses a webcam on a specific skin area to record the symptoms of a particular case, which may then be transferred to our neural network system when it is operative.

## 3. Input Design Data and Optimization

The diagnostics service’s set of data includes the following elements: vital signs of the patient, vocal complaints from the physician, demographic details about the physician, and the existence of characteristic symptoms. Each module’s constituents act as an input parameter to the system. Thus, the adaptive intelligence network’s job is to establish a connection between the participant’s appearance and heart rhythm using self-reported complaints [30]. While health records and vitals are critical in determining the cause of an illness, it has been found that clients’ concerns offer more insights into clinical diagnosis prediction. However, certain skin diseases need a high degree of accuracy from scanning equipment (NSC, 2005). The objective is to develop automatic information systems capable of collecting the data as input and using them to make an accurate diagnosis. The symptom’s input parameters were set to link a specific complaint and a potential skin condition. The integration of the LSTM with the MobileNet V2 is shown in this section through an architectural diagram. MobileNet V2 is used to identify the kind of skin illness, and LSTM is utilized to improve the model’s performance by retaining state information on the characteristics seen in earlier picture categorization generations [31].

### Importance of MobileNet V2 and LSTM

Along with the ReLu element, the MobileNet V2 design includes a residual layer with a stride of 1 and a shrinking layer with a stride of 2. Figure 4 illustrates the structure of the design. Each residue and shrinking layer consists of three sub-layers. The very first layer is the 1 × 1 convolution using ReLu6. Then, depth-wise convolution is the architect’s second level [32]. The depth-wise layer augments the image with a single convolutional layer that performs a simple filtering operation. The third layer in the proposed system is a nonlinear convolution layer. In the output domain of the third layer, the ReLu6 component is utilized. ReLu6 is often used to guarantee the model’s resilience in low-precision circumstances and to improve the model’s unpredictability [33].

Each layer has an exact number of power channels in that general pattern. A filter of size 33 is often employed in current architectural systems, but dropout and batch normalization are used during the training stage. A residual element facilitates flow throughout the system via batch systems, while ReLu6 serves as the activation component. The LSTM is a widely utilized constituent in recurrent neural network designs. It can rely on its learning sequence to solve issues involving patterns prediction. Memory is maintained by memory blocks that include an input and outlet gate, a forgetting gate, and a screen link [34]. These memory cells are all contained inside the abstract LSTM layer module. In grouping MobileNet V2 and LSTM, there is no necessity for acceptable tuning parameters. The following monitoring cases include the rate of learning and bias weights present at both input and output margins. This amalgamation process is much similar to backpropagation neural networks as during the developing process; that is, the weights will be reduced, thus reducing the complexity compared to MobileNet V1. For the period of this arrangement, a gated cell is arranged where the presence and absence of cells are indicated using binary values of ‘0’ and ‘1’. Even at the intermediate stage, the entire state of the cells can be improved, thus adding new information at a particular period using the multiplication process. Moreover, a new vector is generated before passing the data to hidden states in which memory is created under working conditions. Due to the advantages above, MobileNet V2 and LSTM are combined and processed in a three-stage sequence order.

Calculations for the persistent abstract LSTM memory capsule’s input signal are given below, where memory is included in the LSTM module and the condition is described as Pt at period t over the input’s previous hidden vector vtII,
(1)Input Gate:αt=σitWiα+γt−1Wγα+cst−1Wcsα+αbias
(2)Output Gate:βt=σitWiβ+γt−1Wγβ+cstWcsβ+βbias
(3)Forget Gate:ft=σ(itWif+γt−1Wγf+cstWcsf+fbias
(4)Cellstate Gate:cst=ft.cst−1+αt.tanγ(itWics+γt−1Wγcs+csbias)
(5)LSTM Outcome:γt=βt.tanγ(cst−1)

From Equations (1)–(5), at time t, the variable it is used as the input to the LSTM block. The weights W_ia_, W_ib_, W_if_, and W_ics_ are individually connected with an input node, an outputs gate, a forget gate, and a cell state gate. W_ga_, W_gb_, and W_gf_ are the hidden recurrent layer’s weights. The model of integration using dermoscopy or clinical pictures is normalized and learned in automated skin disease detection by image segmentation, extraction of features, and place of the original, and then saved in the training databases [35]. Hybrid segmented extraction and classification techniques are employed to identify skin disorders effectively. The matching algorithm unit tests the characteristics needed to categorize skin disorders. The process of extracting features from testing pictures is linked with extracting features from previously learned images [36,37,38,39,40,41,42,43,44]. The matching algorithm is designed so that if illnesses are identified, they are categorized, and, ultimately, patients receive an e-prescription; if no diseases are found, the system classifies the skin as healthy. Through the treatment regimen, clients obtain related information such as e-prescriptions. Figure 5 illustrates the framework for the skin disease detection system described above.

To assess the performance of the proposed model, testing is conducted via the supplementary computers during the model’s eat performance. The assessment is based on the multitude of times the proposed model correctly classifies the skin illness as a ‘true positive’ and properly classifies the image as not about that particular skin group as a ‘false neg’ [4]. The high false rate refers to the number of times the suggested framework correctly detects the disease. The amount of times the proposed model misunderstands the skin condition is called the ‘wrong neg’. The values obtained of ‘features of a given, “realneg”, falsified optimistic,’ and ‘systematic error’ are used to determine the proposed model’s susceptibility, specificity, and validity. The percentage of correct classification is used to determine the overall efficiency (true positives and negatives).
(6)Accuracy=(TP + TN)(TP + TN + FP +FN)

The recall is the proportion of accurate positive categorization (true positives) among favorable occurrences.
(7)Recall = TPTP + FN 

Precision is the proportion of favorable cases’ accurate positive (positive result) categorization.
(8)Precision = TPTP + FP 

An F measure is known as the harmonics weighted means for accuracy and retrieval.
(9)twoF−Measure =2×Precision× Recall(Precision+Recall)

The primary reason for adopting MobileNetV2 is that the same classification can be achieved in automatic mode with the preprocessing technique even across the entire latency spectrum. Compared with other lightweight models, only 30 percent of parametric images can be added, thus increasing the parametric count to 80 percent of accurate results. As extensions are present in version 2, residual contractions are made with a third layer interaction process that can be projected for entire images with different matrix types.

## 4. Discussion

The performance of our suggested method and other similar approaches is summarized in performance measures table in terms of recall, precision, accuracy, and F-measure. The MobileNet-based models performed much better at identifying the area of interest with little computing effort, and the MobileNet V2 model performed optimally for illness classification. The MobileNet V2 model included LSTM, which affects critical factors such as learning rates and input and output gates, resulting in a more accurate result. When the findings from performance measures table are shown in Figure 6, it is clear that the proposed MobileNet V2–LSTM method outperforms existing state-of-the-art models in almost all performance areas.

### 4.1. Training Dataset

In this kind of deep learning process, a preprocessing dataset is integrated using an accelerometer and gyroscope in the presence of noise filters. Then, input data will be divided into different data points, and, in this stage, half of the data will be concurred, thus resulting in a multi-scale resolution image. Even with such overlapped ends, it can be observed that 128 different datasets are changed concerning body motion. Therefore, a separate conditional study with 15 subjects and 7 training sets is added with input bias. A sample of the dataset for creating input parameters is given in Table 2.

From Table 2, it can be observed that the reference output will be automatically compared with boundary conditions. If any sample value exceeds the original gated values, then the binary value of 0 will be indicated with errors. If limits are satisfied, one will be represented at the user end without any error values. In addition, the look back condition of the abovementioned dataset can be fed as the input using the following command, *defcreate_dataset*(*dataset, look_back* = 1)*:*

It can be observed that in Table 3, performance measures are compared with six different methods in which highly accurate results are observed in MobileNet V2–LSTM, which is greater than 86%. However, other lightweight CNN methods can achieve below 85% due to high errors present in the preprocessing techniques. Moreover, a series of real-time experimentations were conducted based on the original training data from an individual, which is observed from the hand position. In the first stage before the deep learning process, images were not modified, as the array of matrix type was varied with the classification mechanism. However, after implementing deep learning using MobileNet V2–LSTM, different array sets were processed according to the cross-validation training set. The accuracy of results was enhanced, and images were appropriately distributed with low error processing and high measurement capability.

### 4.2. Progress of the Disease Growth

The sustainability of the suggested method may be evaluated by calculating the mean of the assessed attribute value. The numbers after the decimal digits indicate the estimated valuation divergence from the facts. The suggested technique has virtually little value compared with the other approaches discussed in this article. Figure 7 depicts the graphs from Table 4, showing the progression of the illness, which would enable more effective therapy for the patients. The model is effective in predicting pathological development progression. The confidence rating indicates the average degree of certainty with which it estimates the disease’s high area. The suggested model is influential in matching the illness class more accurately with little computational work.

### 4.3. Execution Time

To evaluate the proposed model’s performance, the validation phase execution time is given in Table 5 and Figure 8 in line with previous research. The suggested model took about 1134 s to train over 20 epochs. The processing time for MobileNet V2 with LSTM did not significantly decrease. Nonetheless, MobileNet V2 outperformed MobileNet V1 in other performance evolution metrics. It can be observed in Table 5 that the execution time, termed as a performance index, is measured and compared with CNN and version 1 of MobileNet. Obviously, due to the preprocessing technique, more time can be saved for the simulation execution model for high array ends. Even maximum iteration periods are much less in the case of MobileNetV2, and it is below 100 s. In contrast, high iteration is achieved above 80 with extensive execution capability for another optimization process. This comparison proves that MobileNetV2 is more practical than existing models.

### 4.4. Preprocessing Factors

Deep learning can be applied to all image processing techniques using the preprocessing technique, as training data will be implemented concerning captured images. In addition, two factors provide a unique pathway for monitoring preprocessed pictures, and it is essential to analyze the following factors compared with the original images.

#### 4.4.1. Incinerate Lightning Module

For detecting skin images, models are scanned using PyTorch, an online platform for the location of different photos. The significant advantage of selecting a lightning module is that images can be processed without any error using natural language, and, in this case, python is used for such cases. Moreover, all complex ideas can be solved using PyTorch as additional features with high interactive models. In addition, the scaling of images can be processed as deep learning assimilates with a proposed model without any change in loop representations. The dataset of the lightning model can be combined using the following class command such as class (dataset) and class (lightning), in which the program can be modeled as


*import torch*



*from torch. Utils.data import data set, data loader*



*class (dataset)*



*class (lightning)*


#### 4.4.2. Viewpoint of Images

From Figure 9, three different viewpoints are analyzed such that skin disease that occurred in the hand path can be monitored only with dehaze images. Compared with original representations, the captured negative samples are removed; thus, a multi-scale image can be detected using different texture classifications. Further classifications are managed with proper threshold limits, thus dividing each area under separate small cluster zones. Therefore, to achieve high accuracy in the images, a fusion segmentation can be applied with additional morphology marks.

## 5. Conclusions

Based on MobileNet V2 and the LSTM method, the suggested model was influential in classifying and detecting skin diseases with little computational resources and effort. The result is encouraging; it has the most fantastic accuracy compared with other techniques. The MobileNet V2 architecture is intended to operate with a stride2-enabled portable device. The model is computationally efficient, and when combined with MobileNet V2, the LSTM module improves prediction accuracy by retaining the last timestamp data. The model would be more resilient if it included information about the current state through weight optimizations.

Additionally, the model was compared with several other traditional models such as CNN and FTNN. As shown in the Results and Discussion section, the proposed model outperformed others in classifying and evaluating the progression of tumor development using texture-based information. The bidirectional LSTM may be used to improve the model’s performance further. However, several flaws must be addressed in future growth. When the model’s accuracy was tested against a set of photos taken under circumstances other than those used during testing, it was drastically reduced to slightly under 80%. Finally, the suggested method is not intended to replace current disease diagnostic technologies but to augment them. Laboratory test findings are always more reliable than visual symptom-based diagnoses, and visual examination alone often complicates early diagnosis.

## Figures and Tables

**Figure 1 life-12-00426-f001:**
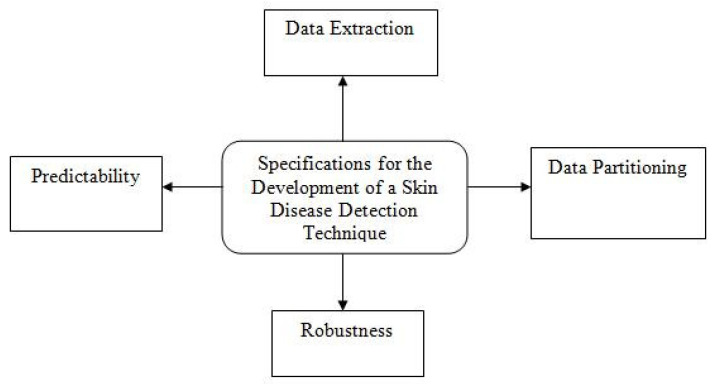
The fundamental criteria for a technique for detecting skin diseases.

**Figure 2 life-12-00426-f002:**
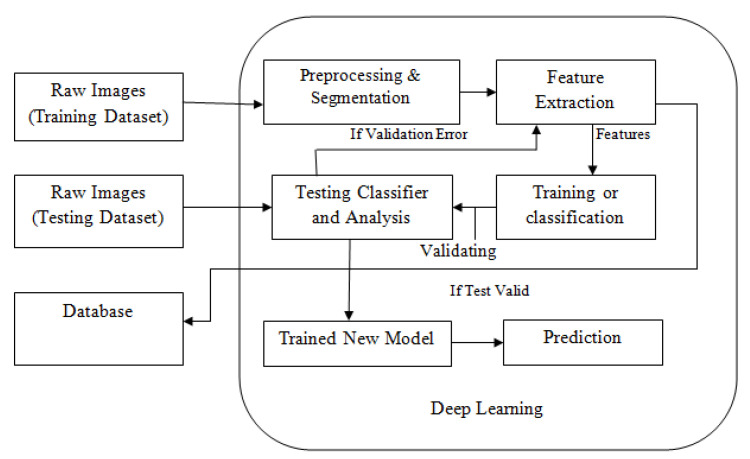
System architecture.

**Figure 3 life-12-00426-f003:**
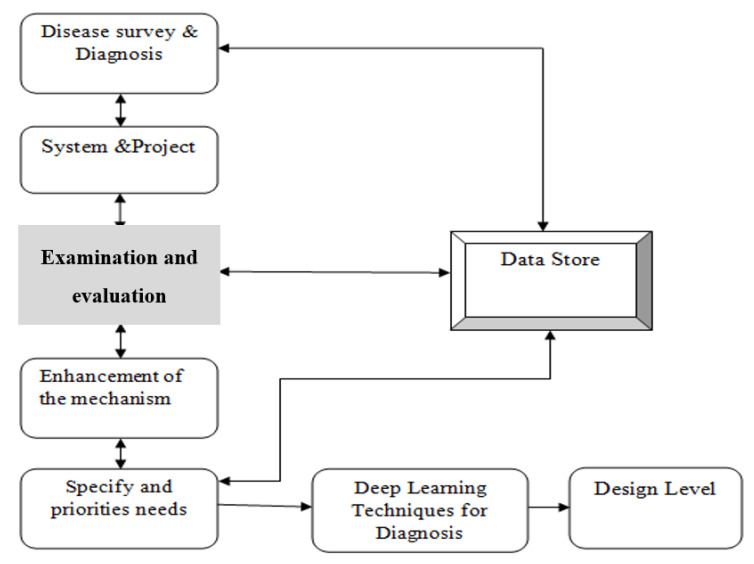
Diagnosis of skin disease.

**Figure 4 life-12-00426-f004:**
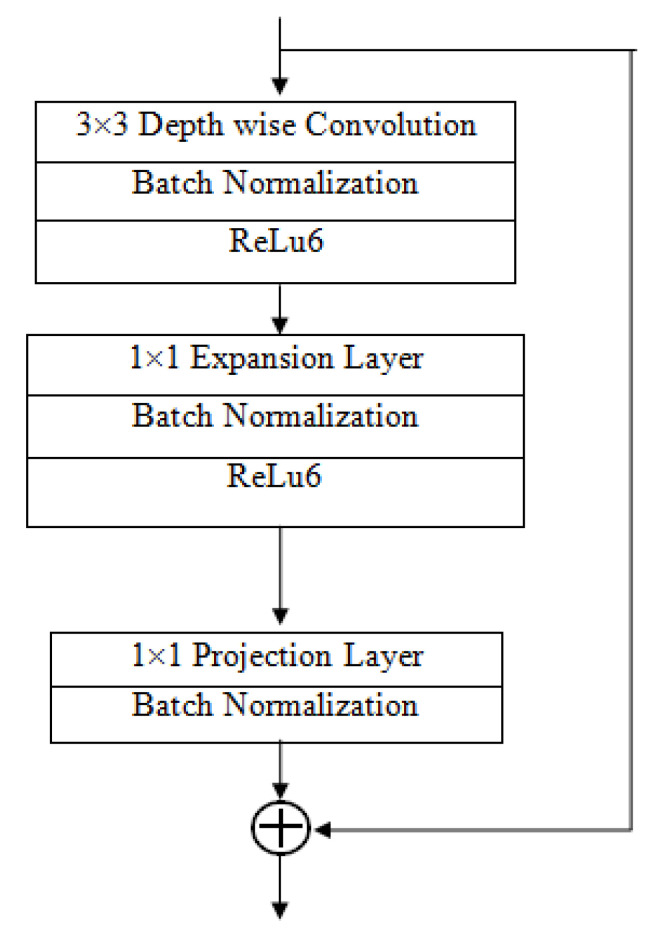
The architecture of MobileNet V2.

**Figure 5 life-12-00426-f005:**
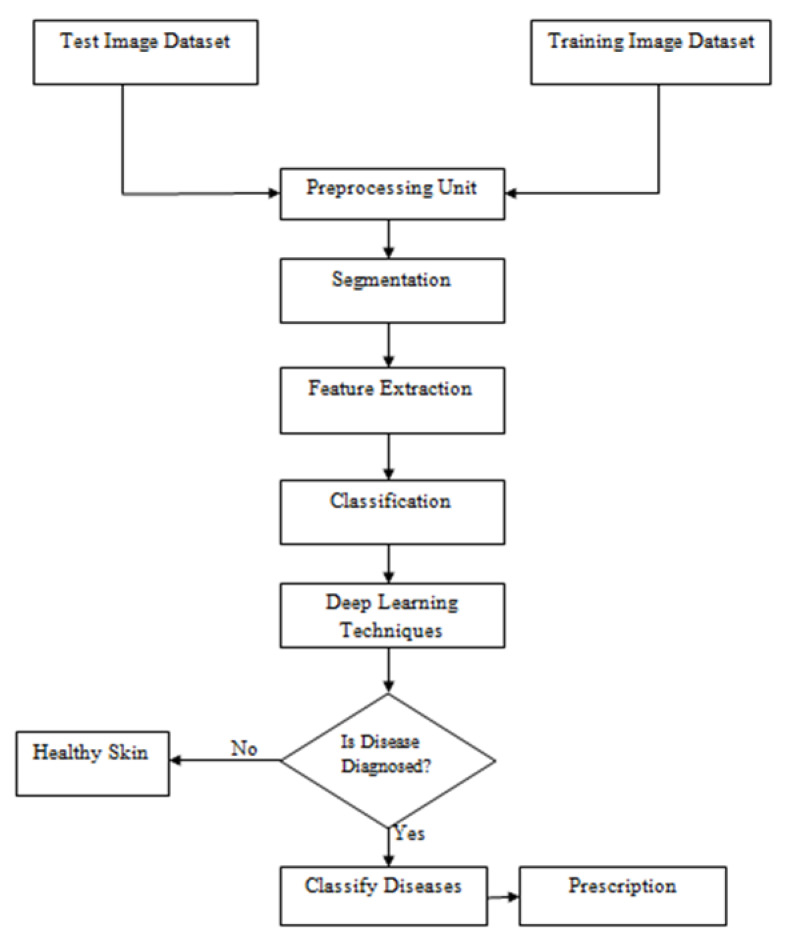
Flowchart for skin disease detection system.

**Figure 6 life-12-00426-f006:**
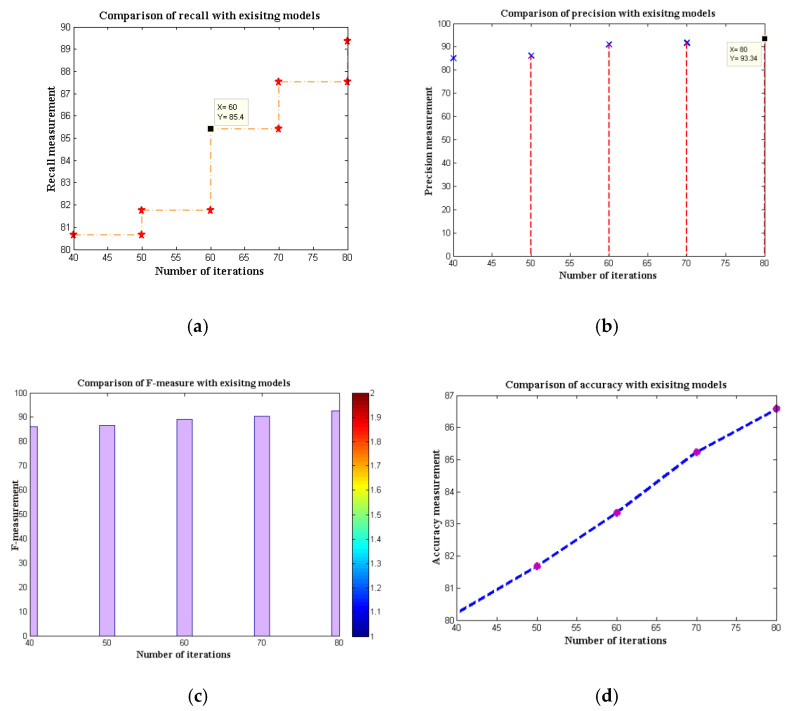
Performance measurement and comparison: (**a**) recall; (**b**) precision; (**c**) F-measure; (**d**) accuracy.

**Figure 7 life-12-00426-f007:**
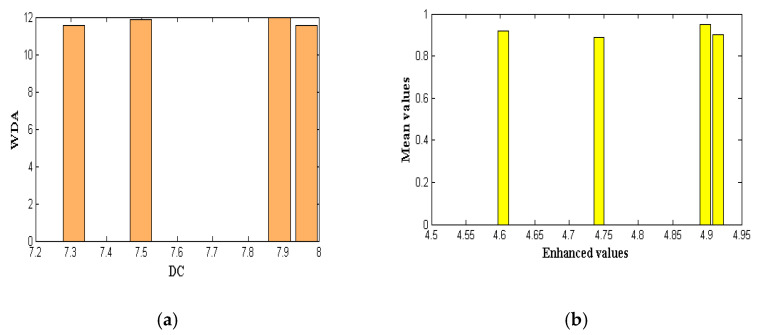
Comparison of MobileNet V2–LSTM (**a**) DC vs. WDA and (**b**) enhanced vs. mean values.

**Figure 8 life-12-00426-f008:**
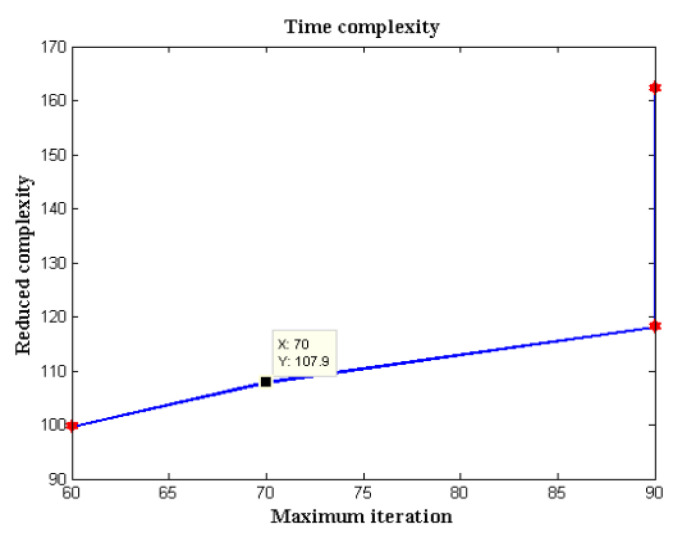
MobileNet V2 execution time using LSTM and other methods.

**Figure 9 life-12-00426-f009:**
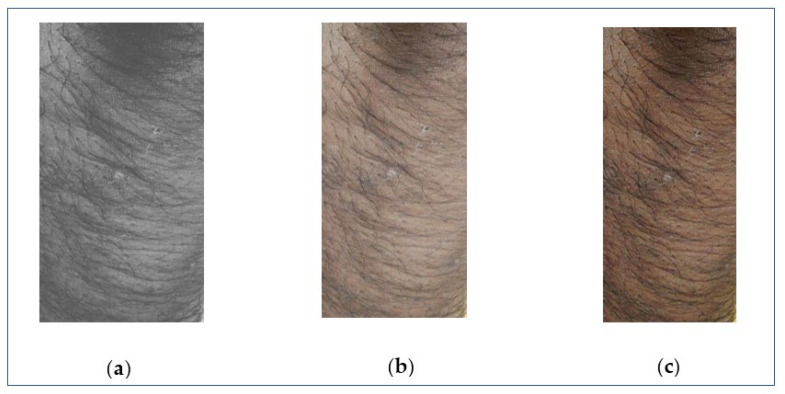
Viewpoint of images: (**a**) negative; (**b**) multi-scale; (**c**) dehaze correlated.

**Table 1 life-12-00426-t001:** Skin diseases and their symptoms.

Diseases	Symptoms
Acne vulgaris	Marks on the face, little pustules, and small lumps
Atopic dermatitis	Itching and irritation, rough, hypersensitivity skin that is genetic, with a red rash and crusty, thick skin
Benign skin tumors	Development that is not harmful, black drawing on the skin, and a smooth, rough, and oily texture
Mastitis	Infectious, heated, and uncomfortable tumor in a woman’s breast
Viral warts	Infection, any region of the organism, exfoliation of the afflicted area
Diaper candidiasis	Fungal, a diaper worn across two legs, urine exposure, red, swelling, seeping, and fluid
Folliculitis	Bacterial, tiny pus vesicles, and little red lumps
Carbuncle	Infection, compromised immune system, diabetes mellitus, and little pimples on the afflicted area
Eczema	Discoloration, cracking, and peeling

**Table 2 life-12-00426-t002:** Input data set using MobileNet V2 and LSTM.

Number of States	Input Gate	Forget Gate	Cell State Gate	Reference Output
1	139	124	144	135.66
2	126	121	128	125
3	143	125	168	145.33
4	159	121	182	154
5	170	147	188	168.33

**Table 3 life-12-00426-t003:** The different methods’ performance measures.

Algorithms	Recall	Precision	F-Measure	Accuracy
FTNN	80.65	85.07	86.07	80.23
CNN	81.75	86.07	86.61	81.67
Depth-based CNN	80.23	80.49	82.56	82.93
Channel boost CNN	81.24	82.39	82.98	83.45
MobileNet V1	85.40	90.92	89.12	83.34
MobileNet V2	87.51	91.69	90.23	85.23
MobileNet V2–LSTM	89.34	93.34	92.68	86.57

**Table 4 life-12-00426-t004:** The progress of disease growth.

Algorithms	Core of Disease (DC)	Whole Disease Area (WDA)	Enhanced	Mean Value
CNN	7.965	11.567	4.743	0.89
Depth-based CNN	7.234	11.459	4.369	0.72
Channel boost CNN	7.348	11.270	4.421	0.81
MobileNet V1	7.309	11.552	4.916	0.90
MobileNet V2	7.498	11.894	4.604	0.92
MobileNet V2–LSTM	7.889	11.999	4.897	0.95

**Table 5 life-12-00426-t005:** Execution time.

Algorithms	Maximum Iteration	Time of Execution
CNN	90	162.32
Depth-based CNN	90	167.90
Channel boost CNN	80	156.23
MobileNet V1	90	118.15
MobileNet V2	70	107.89
MobileNet V2–LSTM	60	99.67

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
