# Peer review of "Deep Learning Approaches for Prognosis of Automated Skin Disease"

_life, 2022, doi:10.3390/life12030426_

Round 1

Reviewer 1 Report

This paper studies the recognition of skin disorders using computer images. CNN and LSTM are used as the classifiers. The topic is important and the paper is well-written in general. However, there are some concerns on the methodology.

1. Further analysis and discussions should be given on the preprocessing of the skin images. Do the factors including lighting, view-point, play an important factor in imaging.

2. Why is MobileNet-V2 is adopted? It should be further compared with other lightweight CNN models. The experimental results are not sufficient.

3. Some exapmles of the correct/incorrect images could be provided and analyzed. Why are they mis-classified and how do the errors reflect on the network structure?

Author Response

Q1. Further analysis and discussions should be given on the preprocessing of the skin images. Do the factors including lighting, view-point, play an important factor in imaging.

Ans: Factors of pre-processing technique is analyzed in revised version of manuscript.

Pre-processing factors

Deep learning can be applied to all image processing technique using pre-processing technique as training data will be implemented with respect to captured images. In addition two factors provides unique pathway for monitoring pre-processed images and it is essential to analyze the following factors in comparison with original images.

Incinerate lightning module

For detecting skin images models are scannedusing pytorch which is an online platform for location of different images. The major advantage of selectinglightning module is that images can be processed without any error using natural language and in this case python is used for such cases. Moreover all complex images can be solved using pytorch as additional features with high interactive models are introduced.Also, scaling of images can be processed as deep learning is assimilated with proposed model without any change of loop representations. The data set of lightning model can be combined using the following class command such as class (dataset) and class (lightning) where the program can be modeled as,

import torch

from torch.utils.data import data set, data loader

class (dataset)

class (lightning)

View point of images

Figure 9. View point of images (a) Negative (b)Multi-scale (c) Dehaze correlated

From Figure 9 it can be observed that three different viewpoints are analyzed where skin disease that occurred in hand path can be clearly monitored only with dehaze images. In comparison with original representations the captured negative samples are removed thus a multi-scale image can be detected using different texture classifications. Further classifications are managed with proper threshold limits thus dividing each area under separate small cluster zones. Thus to achieve high accuracy of images a fusion segmentation can be applied with additional morphology marks.

Included in Pages 12 and 13 of the revised manuscript

---------------------------------------------------------------------------------------------------------------------------------------

Q2. Why is MobileNet-V2 is adopted? It should be further compared with other lightweight CNN models. The experimental results are not sufficient.

Ans: Experimental results are added, compared with two additional methods using Depth based CNN and Channel boost CNN.

The major reason for adopting MobileNet-V2 is that with pre-processing technique same classification can be achieved in automatic mode even across entire latency spectrum. As compared to other lightweight models only 30 percent of parametric images can be added thus increasing the parametric count to 80 percent of accurate results.As extensions are present in version 2 connections are made using residual contractions with third layer interaction process that can be projected for entire images with different matrix type.

Table 2. The different methods' performance measures.

Algorithms

Recall

Precision

F-measure

Accuracy

FTNN

80.65

85.07

86.07

80.23

CNN

81.75

86.07

86.61

81.67

Depth based CNN

80.23

80.49

82.56

82.93

Channel boost CNN

81.24

82.39

82.98

83.45

MobileNet V1

85.40

90.92

89.12

83.34

MobileNet V2

87.51

91.69

90.23

85.23

MobileNet V2-LSTM

89.34

93.34

92.68

86.57

Table 3. The progress of the disease growth.

Algorithms

Core of Disease

(DC)

Whole Disease Area

(WDA)

Enhanced

Mean

Value

CNN

7.965

11.567

4.743

0.89

Depth based CNN

7.234

11.459

4.369

0.72

Channel boost CNN

7.348

11.270

4.421

0.81

MobileNet V1

7.309

11.552

4.916

0.90

MobileNet V2

7.498

11.894

4.604

0.92

MobileNet V2-LSTM

7.889

11.999

4.897

0.95

Table 4. Execution Time.

Algorithms

Maximum iteration

Time of Execution

CNN

90

162.32

Depth based CNN

90

167.90

Channel boost CNN

80

156.23

MobileNet V1

90

118.15

MobileNet V2

70

107.89

MobileNet V2-LSTM

60

99.67

Included in Pages 9,10,11 and 12 of the revised manuscript

---------------------------------------------------------------------------------------------------------------------------------------

Q3. Some exapmles of the correct/incorrect images could be provided and analyzed. Why are they mis-classified and how do the errors reflect on the network structure?

Ans: Examples of correct/Incorrect images are added in revised version of manuscript.

View point of images

 Figure 9. View point of images (a) Negative (b) Multi-scale (c) Dehaze correlated

From Figure 9 it can be observed that three different viewpoints are analyzed where skin disease that occurred in hand path can be clearly monitored only with dehaze images. In comparison with original representations the captured negative samples are removed thus a multi-scale image can be detected using different texture classifications. Further classifications are managed with proper threshold limits thus dividing each area under separate small cluster zones. Thus to achieve high accuracy of images a fusion segmentation can be applied with additional morphology marks.

Included in Page 13 of the revised manuscript

---------------------------------------------------------------------------------------------------------------------------------------

Reviewer 2 Report

This paper describes a system for the automatic identification of skin conditions. The authors start by describing the problems related to the skin in the introduction, which is too long and unnecessary. The authors should spend more time explaining the solution, and not so much about the problem itself. In subsection 1.2 authors should have explained the motivation for their work, however, instead they techniques and a diagram that does not contribute to the quality of their presentation. Reading the introduction does not help in understanding the contribution of the paper. 

Section 2 and Section 3 also poorly describe the model of the system. It is unclear what is the motivation for combining the MobileNet and LSTM. Figures 2 and 3 are not described well in this paper. 

Overall, the paper is very challenging to read due to its poor organization and extensive errors in writing. The paper needs to be significantly revised in order to be reviewed again. The abstract should be revised too.

It is unclear what types of data were used for training and testing.

These are only some of the sentences that remain unclear:

  1. Approaches to a thorough investigation It is possible that deep learning methods might be used to construct frameworks capable of identifying diverse skin conditions.
  2. The human body's primary feature is the skin. The epidermis, dermis, and subcutaneous tissues comprise it.
  3. Skin disorders are a significant impact on one's life and the well.
  4. The technological development allows for the structuring and execution of a skin observation early years foundation identification of skin problems.
  5. Infections may be diagnosed using image classification and deep learning [3].
  6. Currently, the skin irritation condition is detected only via biopsies at a future date.
  7. Skin disease classification based on the picture of the afflicted area suggested work.
  8. The woman's actual use is to develop an app that captures a picture of the afflicted area of the skin in order to classify it [8].
  9. The scientists chose the MobileNet V2 model because it...
  10. To get the binary mask of a skin lesion, standard segmentation methods are combined with a fusing approach.
  11. With the this classification algorithm in place, the program can now estimate the illness for a fresh picture of a skin condition received through in an Android mobile application [14].
  12. Figure 2 depicts the proposed program's design, which depicts the proposed system's major operations.
  13. Image pre-processing is a critical stage in the detection method since it allows for the noise removal such as hairs, clothes, as well as other artifacts while also improving the characteristics of the original image [16].
  14. This session will analyze these illnesses [25].
  15. If a result is made, the physician suggests the illness as a potential cause of skin disease.
  16. Thus, the professional is heavily reliant on his expertise and the assessment of the client complaints.
  17. The phases of this study are shown in Figure 3 [27]. Figure 3. Dermatological Disease Analyses of Diagnostic Systems Figure 3 illustrates the illness assessment procedure, which comprises the basic gathering and cataloguing of skin disorders that are researched and utilized in the design process.
  18. The assessment is based on the multitude of times the proposed model correctly classifies the skin illness...

Author Response

Q1. This paper describes a system for the automatic identification of skin conditions. The authors start by describing the problems related to the skin in the introduction, which is too long and unnecessary. The authors should spend more time explaining the solution, and not so much about the problem itself.In subsection 1.2 authors should have explained the motivation for their work, however, instead they techniques and a diagram that does not contribute to the quality of their presentation. Reading the introduction does not help in understanding the contribution of the paper.

Ans: Solutions for avoiding problems and motivation of proposed work is described in revised version of article.

Drawbacks of conventional method: A solution

In all the above mentioned process researchers have contributed the process of identifying skin disease using various methods such as computational models, diagnostic techniques and morphological procedures. However, such models are not sufficient for correlating skin diseases for long period of time. Thus a new solution can be provided using a deep learning technique where point of images will be classified based on three different images. Typically in other procedures original images are processed and compared with correlated ones but in deep learning process if skin infections are found then original images will be dehazed and negative impression of same image can be obtained thus achieving real time operating results with clear view of comparison.

Motivation of proposed work

The major impetusof projected work is that presence of skin diseases can be identified at early stages thus preventing serious observation in each individual. Moreover a computer based detection process is also designed and fabricated for extracting several feature set even if data is much higher. Since data is long other methods can able to provide correlated image point after some period of time but motivation of implementing deep learning procedures in testing stages can able to predict incorrect values and images which is not possible in other computational models. Further the major reason for integrating mobile net V2 with LSTM provides great advantage on extracting static images as many indoor positions will be detected at discrete points of the body. In case if positions and postures are changing then mobile net V2 with LSTM provides high accuracy even with supplied low resource modules.

Included in Page 4of the revised manuscript

---------------------------------------------------------------------------------------------------------------------------------------

Q2. Section 2 and Section 3 also poorly describe the model of the system. It is unclear what is the motivation for combining the MobileNet and LSTM. Figures 2 and 3 are not described well in this paper.

Ans: Motivation for combining MobileNet and LSTM with explanation of Figures 2 and 3 are added.

Motivation of proposed work

The major impetus of projected work is that presence of skin diseases can be identified at early stages thus preventing serious observation in each individual. Moreover a computer based detection process is also designed and fabricated for extracting several feature set even if data is much higher. Since data is long other methods can able to provide correlated image point after some period of time but motivation of implementing deep learning procedures in testing stages can able to predict incorrect values and images which is not possible in other computational models. Further the major reason for integrating mobile net V2 with LSTM provides great advantage on extracting static images as many indoor positions will be detected at discrete points of the body. In case if positions and postures are changing then mobile net V2 with LSTM provides high accuracy even with supplied low resource modules.

Figure 2 illustrates the architecture on discrete data processing at stage 1 as raw images are divided with a two period data set and database. Since two stages are defined a correlated data is established that allows pre-processing and segmentation at stage 2 with flexible error validation procedures. However the divided data set will be used for classifying and analyzing skin diseases without any validation procedures at stage 3. Both stage 2 and 3 are combined for training a new model in case of test results is valid for certain period of time. At final stage of deep learning model a feature extraction and prediction is managed thus detecting abnormal changes in the skin.

Figure 3 enhances the description of abovementioned system architecture after detecting the abnormal data as diagnosis process will be carried forward with storage mechanisms. Therefore, the monitored data will be separated as standard and nonstandardprocedures thus allowing the system to examine the proposed design with pre-existing individuals. Again at this intermediate stage data will be re-stored and 50% of abnormalities in images will be removed. This removal of abnormality procedure is followed by enhancement mechanism which allows the priorities to be diagnosed at design state.

Included in Pages 4,5 and 6of the revised manuscript

---------------------------------------------------------------------------------------------------------------------------------------

Q3. Overall, the paper is very challenging to read due to its poor organization and extensive errors in writing. The paper needs to be significantly revised in order to be reviewed again. The abstract should be revised too.

Ans: Abstract is revised and errors are modified in the revised version of manuscript.

In current generation most of the individuals are encrustation distress due todifferent skin problems and ailments that spreads all around the world. Regardless of its spreading rate, the method of assessing skin diseases is very difficult due to high complexities such as skin tones, hair colors, and hairstyles. A high disorder that is found in individual indicatesthat a significant public health risk is present across the globe and they become more dangerous when they enter the invasive phase. In addition increased pollution and poor diet maintenance increases the number of skin disorders at an alarming rate as early signs of skin illness are often overlooked by many people. Thus the projected approach is implemented for diagnosing skin diseases that depends ona tissue removal as examined and administered by physicians. With the hybrid technique, a circumventprocedure is implemented with human assessment and error diagnoses marks in pre-processing images at appropriate period of time. Approaches to a systematic investigation indicate that deep learning methods can be used to construct frameworks where diverse skin conditions can be identified.Skin and non-skin tissue must be distinguished in order to detect skin diseases thus a classification system has been developed using MobileNet-V2 and LSTM. For the proposed system modelprimary objective is to ensure accuracy in skin disease forecasting which greatly increases the efficiency in storing high state information for exact forecasts.

Included in Page 1of the revised manuscript

---------------------------------------------------------------------------------------------------------------------------------------

Q4. It is unclear what types of data were used for training and testing.

Ans: Data for training and testing is illustrated in Figure 9.

View point of images

Figure 9. View point of images (a) Negative (b) Multi-scale (c) Dehaze correlated

From Figure 9 it can be observed that three different viewpoints are analyzed where skin disease that occurred in hand path can be clearly monitored only with dehaze images. In comparison with original representations the captured negative samples are removed thus a multi-scale image can be detected using different texture classifications. Further classifications are managed with proper threshold limits thus dividing each area under separate small cluster zones. Thus to achieve high accuracy of images a fusion segmentation can be applied with additional morphology marks.

Included in Pages 13 and 14of the revised manuscript

---------------------------------------------------------------------------------------------------------------------------------------

Q5. These are only some of the sentences that remain unclear.

Ans: As per reviewer comments sentences are modified in revised version of manuscript.

  1. Approaches to a systematic investigation indicate that deep learning methods can be used to construct frameworks where diverse skin conditions can be identified.
  2. The human body primary feature is the skin that comprises of the epidermis, dermis, and subcutaneous tissues.
  3. Skin disorders create a significant impact on individual human life.
  4. The technological development in detection of skin disease allows different structuring and execution of skin observationsin early year foundations.
  5. Infections can be diagnosed using image classification and deep learning [3].
  6. In current generation systems many skin irritation problems are detected only usingbiopsies which allow operating of individual membraneat a future date.
  7. In the projected workskin disease classification is based on the correlated images onafflicted areas.
  8. The major reason of such developments is to use the applicationin real time that captures a picture of the afflicted area of the skin in order to classify it [8].
  9. Most of the researchers choose MobileNet V2 model because it is computationally efficient and performs well with limited quality pictures.
  10. To get the binary values in deep learning process during wound process, a standard segmentation method is used and combined with a fusing approach.
  11. With this classification algorithm the combined model can estimate the illness of an individual with a fresh picture of individual skin that is received through developed application platform [14].
  12. Figure 2 depicts the proposed design parameters that are used for executing all major operations in the system.
  13. Image pre-processing is a critical stage in skin detection method as it allows removal of unwanted quantities such as hairs, clothes etc.This category of image pre-processing is used for improving the characteristics of the original image [16].
  14. Thus this section analyzes the importance of illnesses [25] and additionally describes the topology of an artificial neural network that is used to diagnose skin disorders.
  15. If diagnoses are made with respect to classified images then physician suggests the illness as a potential cause of skin disease.
  16. Thus, many skin specialistsdepend on their expertise and the assessment of the client complaints.
  17. The step-by-step stages of the proposed work is shown in Figure 3 where the illness assessment procedures are discussed that comprises the basic gathering and cataloguing of skin disorders that are researched and utilized in the design process.
  18. The assessment is based on simulation time periods that allows the proposed model to classify the skin illness in two different approaches such as a True Positive which properly classifies the image andsubsequent non-pertaining classifiers to the particular skin group as False Neg [4].

Included in Page 1,2,3,5,6 and 10of the revised manuscript

---------------------------------------------------------------------------------------------------------------------------------------

Reviewer 3 Report

The paper presents an effective way to identify and treat skin diseases. It is a topic of interest to the researchers in the related areas. For the reader, however, a number of points need clarifying and certain statements require further justification. My detailed comments are as follows:

(1)Is the accuracy of the defined target classification dataset based in this paper, so as to ensure the accuracy of image classification diagnosis of skin diseases?

(2)Does the greatly reduced accuracy of the diagnostic results obtained in different cases indicate that the practicality of this model is unable to meet the social needs?

(3)There is no significant difference in execution time in Section 4.2. Please briefly describe the performance index that MobileNet V2 is better than MobileNet V1?

(4)It is noteworthy that your paper needs to be carefully edited in that format.For example: the center format of the article illustrations, the formula editor writing formula, and so on.

Author Response

Q1. Is the accuracy of the defined target classification dataset based in this paper, so as to ensure the accuracy of image classification diagnosis of skin diseases?

Ans: Point of image classification has been added in revised version of manuscript.

View point of images

Figure 9. View point of images (a) Negative (b) Multi-scale (c) Dehaze correlated

From Figure 9 it can be observed that three different viewpoints are analyzed where skin disease that occurred in hand path can be clearly monitored only with dehaze images. In comparison with original representations the captured negative samples are removed thus a multi-scale image can be detected using different texture classifications. Further classifications are managed with proper threshold limits thus dividing each area under separate small cluster zones. Thus to achieve high accuracy of images a fusion segmentation can be applied with additional morphology marks.

Included in Page 13 of the revised manuscript

---------------------------------------------------------------------------------------------------------------------------------------

Q2. Does the greatly reduced accuracy of the diagnostic results obtained in different cases indicate that the practicality of this model is unable to meet the social needs?

Ans: Accuracy of proposed model is compared and discussed.

Table 2. The different methods' performance measures.

Algorithms

Recall

Precision

F-measure

Accuracy

FTNN

80.65

85.07

86.07

80.23

CNN

81.75

86.07

86.61

81.67

Depth based CNN

80.23

80.49

82.56

82.93

Channel boost CNN

81.24

82.39

82.98

83.45

MobileNet V1

85.40

90.92

89.12

83.34

MobileNet V2

87.51

91.69

90.23

85.23

MobileNet V2-LSTM

89.34

93.34

92.68

86.57

It can be observed that in Table 2 performance measures are compared with six different methods where high accurate results are observed in MobileNet V2-LSTM which is greater than 86%. However other lightweight methods using CNN can able to achieve below 85% due to high errors that are present in pre-processing techniques.Also series of real time experimentation has been conducted based on original training data from an individual which is observed from hand position. In first stage before deep learning process images are not modified as array of matrix type is varied with classification mechanism. However after implementation of deep learning using MobileNet V2-LSTMeven different array set is processed according to cross-validation training set. Thus accuracy of results are enhanced and images are properly distributed with low error processing and high measurement capability.

Included in Pages 10 and 11 of the revised manuscript

---------------------------------------------------------------------------------------------------------------------------------------

Q3. There is no significant difference in execution time in Section 4.2. Please briefly describe the performance index that MobileNet V2 is better than MobileNet V1?

Ans: Comparison analysis has been described in brief.

Table 4. Execution Time.

Algorithms

Maximum iteration

Time of Execution

CNN

90

162.32

Depth based CNN

90

167.90

Channel boost CNN

80

156.23

MobileNet V1

90

118.15

MobileNet V2

70

107.89

MobileNet V2-LSTM

60

99.67

It can be observed that in Table 4 that execution time which is termed as performance index is measured and compared with CNN and Version 1 of MobileNet. It is obvious that due to the presence of pre-processing technique more amount of time can be saved for simulation execution model for high array ends. Even maximum iteration periods are much lesser in case of MobileNet Version 2 and it is below 100 seconds whereas for other optimization process high iteration is achieved above 80 with large execution capability. This comparison proves that MobileNet Version 2 is much effective as compared with existing models.

Included in Page 12 of the revised manuscript

---------------------------------------------------------------------------------------------------------------------------------------

Q4. It is noteworthy that your paper needs to be carefully edited in that format. For example: the center format of the article illustrations, the formula editor writing formula, and so on.

Ans: As per reviewer comment paper has been edited with Format of Life MDPI.

---------------------------------------------------------------------------------------------------------------------------------------

Round 2

Reviewer 1 Report

The authors have addressed my previous concerns in the revised version. Error cases have been further discussed. Detailed comparison between models have been addes. 

Author Response

Manuscript ID: life-1606121

Title:  DEEP LEARNING APPROACHES FOR PROGNOSIS OF AUTOMATED SKIN DISEASE

Authors: Pravin R Kshirsagar, Hariprasath Manoharan, Shitharth S, Abdulrhman, M. Alshareef, NabeelAlbishry, Praveen Kumar Balachandran

The Authors thank the discussers for their keen interest in our work, and for their comments.

We have the following reply offer to the discussers’ Comments:

Reviewer (1)

Q1. The authors have addressed my previous concerns in the revised version. Error cases have been further discussed. Detailed comparison between models have been added.

Ans: The authors thank the reviewer for his valuable suggestions.

--------------------------------------------------------------------------------------------------------------------------------------

Reviewer 2 Report

Unfortunately, I cannot recommend this paper for publishing. After carefully reviewing the second version of the manuscript I have detected many faults that have not been taken into consideration during revision. Even the revised text has many grammatical errors.

The motivation for combining MobileNet and LTSM remains unclear, as does the important information about the dataset that was used for training/testing. 

The paper is still very challenging to read due to its poor organization and extensive errors in writing. 

Author Response

Manuscript ID: life-1606121

Title:  DEEP LEARNING APPROACHES FOR PROGNOSIS OF AUTOMATED SKIN DISEASE

Authors: Pravin R Kshirsagar, Hariprasath Manoharan, Shitharth S, Abdulrhman, M. Alshareef, NabeelAlbishry, Praveen Kumar Balachandran

The Authors thank the discussers for their keen interest in our work, and for their comments.

 We have the following reply offer to the discussers’ Comments:

Reviewer (2)

Q1. Even the revised text has many grammatical errors.

Ans: As per reviewer comment grammatical errors have been checked using Grammar.ly and modified.

---------------------------------------------------------------------------------------------------------------------------------------

Q2. The motivation for combining MobileNet and LTSM remains unclear, as does the important information about the dataset that was used for training/testing.

Ans: Motivation for combining MobileNet and LSTM with training dataset has been added.

Ingrouping of MobileNet V2 and LSTM there is no necessity for fine tuning of parameters as the following monitoring cases such as rate of learning, bias weights that are present at both input and output margins. This process of amalgamation is much similar to back propagation neural networks as during developing process weights will be reduced thus reducing the complexity as compared to MobileNet V1. For the period of this arrangementa gated cell is arranged where presence and absence of cells are indicated using binary values of ‘0’ and ‘1’. Even at intermediate stage entire state of the cells can be improvedthus adding new information at particular period of time using multiplication process. Moreover a new vector is generatedbefore passing the information to hidden states where a memory is created under working conditions. Due to the aforementioned advantages both MobileNet V2 and LSTM are combined and information is processed in a three stage sequence order.

Training dataset

In this kind of deep learning process a pre-processing data set is integrated using accelerometerand gyroscopein the presence of noise filters. Then input data will be divided with different data pointsand in this stage half of the data will be concurred thus giving a multi-scale resolution image. Even with such overlapped points it can be observed that 128 different data set is changed with respect to motion of body. Thus in this case a separate conditional study with 15 subjects and 7 training set is added with input bias. A sample of data set for creating input parameters is given in Table 2.

Table 2 Input data set using MobileNet V2 and LSTM

Number of states

Input gate

Forget gate

Cell state gate

Reference output

1

139

124

144

135.66

2

126

121

128

125

3

143

125

168

145.33

4

159

121

182

154

5

170

147

188

168.33

From Table 1 it can be observed that reference output will be automatically compared with boundary conditions and if any sample value exceeds the original gated values then binary value of 0 will be indicated with errors. If limits are satisfied then 1 will be represented at user end without any error values. Also the look back condition of abovementioned data set can be fed as input using the following command.

defcreate_dataset(dataset, look_back=1):

Included in Pages 7 and 10of the revised manuscript

---------------------------------------------------------------------------------------------------------------------------------------

Q3. The paper is still very challenging to read due to its poor organization and extensive errors in writing.

Ans: Paper has been organized in correct way and errors in writing are checked and reduced.

Organization of paper

Abstract

  1. Introduction
    • Literature review
    • Research gap
  2. System model
    • Motivation of proposed work
  3. Design data
    • Importance of MobileNet V2 and LSTM
  4. Results and discussions
    • Progress of the disease growth
    • Execution time
    • Pre-processing factors
      • Incinerate lightning module
      • View point of images
  1. Conclusions

---------------------------------------------------------------------------------------------------------------------------------------
